# High Prevalence of Clonal Reproduction and Low Genetic Diversity in *Scutellaria floridana*, a Federally Threatened Florida-Endemic Mint

**DOI:** 10.3390/plants12040919

**Published:** 2023-02-17

**Authors:** Gina Renee Hanko, Maria Therese Vogel, Vivian Negrón-Ortiz, Richard C. Moore

**Affiliations:** 1Department of Biology, Miami University, Oxford, OH 45056, USA; 2Florida Ecological Services Field Office, U.S. Fish and Wildlife Service, 1601 Balboa Ave., Panama City, FL 32405, USA

**Keywords:** plant conservation genomics, *Scutellaria floridana*, clonality, guerrilla strategy, managed lands

## Abstract

The threatened mint Florida skullcap (*Scutellaria floridana*) is endemic to four counties in the Florida panhandle. Because development and habitat modification extirpated several historical occurrences, only 19 remain to date. To inform conservation management and delisting decisions, a comprehensive investigation of the genetic diversity and relatedness, population structure, and clonal diversity was conducted using SNP data generated by ddRAD. Compared with other Lamiaceae, we detected low genetic diversity (*H_E_* = 0.125–0.145), low to moderate evidence of inbreeding (*F*_IS_ = −0.02–0.555), and moderate divergence (*F*_ST_ = 0.05–0.15). We identified eight populations with most of the genetic diversity, which should be protected in situ, and four populations with low genetic diversity and high clonality. Clonal reproduction in our circular plots and in 92% of the sites examined was substantial, with average clonal richness of 0.07 and 0.59, respectively. *Scutellaria floridana* appears to have experienced a continued decline in the number of extant populations since its listing under the Endangered Species Act; still, the combination of sexual and asexual reproduction may be advantageous for maintaining the viability of extant populations. However, the species will likely require ongoing monitoring, management, and increased public awareness to ensure its survival and effectively conserve its genetic diversity.

## 1. Introduction

Genetic diversity is a fundamental component of biodiversity and has profound effects on ecological processes, including community structure and the ability of populations to recover from disturbance. Research in the field of conservation genetics has shown that loss of genetic diversity due to inbreeding reduces the ability of a population to persist in unstable environments [1]. Despite its widely recognized importance, the implementation of conservation genetics in management is lagging behind research efforts [2,3,4,5,6]. Information on inbreeding, population divergence, and effective population size has the potential to greatly increase the efficacy of conservation practices such as population viability analysis and adaptive habitat management [7,8].

Understanding genetic factors such as diversity, reproductive strategies, and gene flow in rare plant species is especially crucial in an era of rapidly changing habitats. Due to their limited migration capabilities and often highly specialized adaptations to soil and moisture conditions, rare plants are especially vulnerable to the impacts of climate change and increasing land conversion due to human activities, and their survival will likely depend on their adaptive abilities [9]. Maximizing genetic diversity in rare plants and understanding the genetic impacts of conservation activities such as habitat restoration, assisted migration, and ex situ propagation are vital to their protection [10,11,12].

One such rare species is the Florida skullcap (*Scutellaria floridana* Chapman, Lamiaceae), a perennial wildflower endemic to four counties in the Florida panhandle (Figure 1 and Figure 2). It grows in fire-dependent habitats such as longleaf pine wet forests and wet meadows and has a strong response to fire, typically flowering from April to December following fires [13]. Development of slash pine plantations has eliminated large areas of suitable habitat, and *S. floridana* was federally listed as threatened in 1992. Ten of the 29 historically documented populations of *S. floridana* have been extirpated due to habitat modification, and the 19 that remain continue to be threatened by urban development, timber farming, and fire suppression [13].

Only one study of genetic diversity in *S. floridana* has been conducted to date. This study used amplified fragment length polymorphism (AFLP) molecular markers to analyze genetic diversity among 197 samples collected from seven populations [14]. The authors reported moderate genetic diversity and low population differentiation. This study provides a basic groundwork for understanding the genetic diversity of the species; however, it is limited by including a small number of populations. In-depth genetic analysis of the remaining populations of *S. floridana* is vital to accurately determine their viability and inform the process of recovery under the Endangered Species Act (ESA).

Surveys have shown increases in the number of stems within several populations on managed lands over the past 10 years, and the majority of the remaining populations have good or excellent estimated viability [13]. However, population trends are poorly understood because plants can spread via underground rhizomes, which makes it difficult to determine how many stems are part of a single ramet rather than whether they represent distinct individuals [13]. The extent of clonal propagation in *S. floridana* represents a significant knowledge gap for the species. In addition, the type of clonal growth strategy, i.e., phalanx (closely spaced ramets with short internodes) or guerrilla (widely spaced ramets with long internodes) [15], is also unknown.

While clonal propagation increases an individual’s chance of reproduction, increases population sizes, enhances longevity in disturbance-prone habitats, and allows reproduction when conditions are unfavorable for flowering or seed germination, it also poses issues for rare species and complicates conservation efforts [16,17]. When clonal propagation becomes the primary reproductive strategy of a population, it can impact genetic variation and effective population size, or the number of reproductively viable individuals present in a population [18,19]. Since the stems that comprise a ramet are genetically identical, they are essentially a single individual for genetic purposes. As a result, a population that experiences high levels of clonal propagation and is composed of only a few genetically distinct individuals, or genets, would have an effective population size that is smaller than the number of stems observed. Additionally, it is often impossible for biologists to identify stems as genets or ramets in the field, leading to overestimations of population size and viability [17].

This study has three objectives to help evaluate the current status of *S. floridana*. These are to (1) investigate changes in land use that resulted in population extirpation over the last 20 years, (2) assess the genetic diversity of several populations across the species’ range, and (3) determine the prevalence of clonality in the species. The conclusions drawn from this study will contribute to the recovery actions of *S. floridana* as specified in its recovery plan [20] and will address the current knowledge gap regarding the prevalence of clonality, which was identified as a significant problem impeding the ability to accurately assess the abundance of *S. floridana* [13]. Our assessment of genetic diversity will allow us to better evaluate the viability of several remaining populations, and our findings regarding the prevalence of clonality will allow us to generate more accurate estimates of effective population size.

## 2. Results

We were able to locate individuals of *S. floridana* in 12 of the 17 accessible populations (Figure 2). Five populations did not have any above-ground specimens of *S. floridana* after prolonged surveying throughout the entirety of the area. Three of these are located along power easements; one is located in a private reserve; and one is located in an empty lot in central Gulf County. While most of these locations contained at least one of the species commonly associated with *S. floridana*, such as *Aristrida stricta* (Wiregrass) and *Sphagnum strictum* (Pale bug-moss), they all lacked an overstory of *Pinus palustris* (Longleaf Pine) and were located less than 100 m from major roadways. We were also unable to locate plants in one population that is located in a small area of St. Joseph Bay State Buffer Preserve (SJSBP, Port St Joe, Gulf Co., USA) to the north of the main preserve. Several of the sites within the population were inaccessible due to fencing, but those that we could access were flooded by approximately 30–50 cm of water with an extremely thick understory of *Cyrilla racemiflora* (Titi) and little to no herb layer. Two of the populations that we attempted to visit were located in active rangeland or timber harvest operations in central Gulf County and were inaccessible. Of the 12 populations that contained actively growing individuals, Box-R Wildlife Management Area population 1 (BRWMA1, Apalachicola, Franklin Co., USA) was the smallest, containing seven small patches (<10 stems/patch and <5 m^2^), while larger populations contained hundreds of stems. At the time of collection, we observed flowering individuals in several sites in the Apalachicola National Forest (ANF, Sumatra, Liberty Co., USA), one site in Tate’s Hell State Forest (THSF, Franklin Co.), one site in the BRWMA, and the only site in the Lathrop Bayou Habitat Management Area (LB, Bay Co., USA).

### 2.1. Spatial Analysis

Between 2001 and 2019, the approximately 309,000 km^2^ area that represents the entirety of *S. floridana*’s range experienced significant increases in medium- and high-intensity development (21% and 14%, respectively). Deciduous and evergreen forests decreased by 33% and 14%, respectively, while mixed forest, shrubland, grassland, and herbaceous wetlands increased by 18%, 19%, 39%, and 22%, respectively. Approximately 22% of *S. floridana*’s range falls within the borders of public lands, including state forests, national forests, habitat management areas, and preserves. These areas experienced no significant change (>2%) in land cover during this time frame, suggesting that changes were restricted to private lands (Appendix A).

Areas within 500 m of the 12 *S. floridana* populations that we confirmed to be extant during our surveys are composed almost entirely of woody wetlands and evergreen forest, which account for over 88% of land cover. Notably, these areas also contain no mixed forest, pasture, or cultivated crops, and low, medium, and high intensity development and deciduous forest each account for <0.5% of land cover. There was minimal change in land cover in these areas between 2001 and 2019 (Figure 3).

Areas within 500 m of the five populations where we failed to find *S. floridana* contained more woody wetlands and less evergreen forest than the areas containing extant populations, with woody wetlands accounting for over 70% of land cover and evergreen forest accounting for 8.5% of land cover. They also contained more developed space than evergreen forest and, unlike the areas that contained extant populations, included high intensity development, pasture, and cultivated crops.

### 2.2. Population Structure and Diversity

The final dataset that we used for downstream analysis included 10,223 loci and 28,210 variable sites that were each present in at least 60% of individuals. The observed heterozygosity ranged from 0.04 to 0.14, with an average value of 0.11 (Table 1). The observed heterozygosity was substantially less than expected for BRWMA1, ANF5, SJSBP1, and THSF1 (ΔH = 0.099, 0.080, 0.055, and 0.050, respectively; Table 1). These populations also displayed high inbreeding coefficients (*F*_IS_ = 0.56, 0.50, 0.27, and 0.30, respectively).

Populations generally displayed moderate divergence from one another (*F*_ST_ = 0.05–0.15). Nine pairs of populations showed little to no differentiation from one another (*F*_ST_ ≤ 0.05), and six pairs of populations showed high divergence from one another (*F*_ST_ = 0.15–0.25). Overall, LB1, SJSBP1, SJSBP2, THSF1, and THSF2 all showed low divergence from one another as a group, with pairwise *F*_ST_ falling at or below 0.07 for all pairs. Five populations, ANF1, ANF2, ANF3, BRWMA2, and THSF2, also had high *F*_ST_ values relative to all other populations and particularly high divergence values (*F*_ST_ > 0.20) relative to ANF5 and BRWMA1 (Figure 4). Within the Apalachicola National Forest, ANF5 showed moderate to high divergence from all other ANF populations despite its geographic proximity (Figure 4). Mantel test detected a significant positive relationship between *F*_ST_ and geographic distance (R^2^ = 0.29, *p* = 0.02).

Our analysis of ancestry proportions identified ten ancestral populations (K = 10) as the most likely scenario. Lathrop Bayou, SJSBP2, BRWMA2, ANF1, ANF2, ANF3, ANF4, and THSF2 all contained individuals possessing unique ancestry proportions that were not present in any other population. The four remaining populations (ANF5, SJSBP1, BRWMA1, and THSF1) were almost entirely composed of individuals with highly admixed ancestry estimates and did not display an identifiably unique pattern of ancestry (Figure 5).

Cross-validation identified the optimal number of PCs to retain as 20. Of the eight possible discriminant axes, all were retained. The first two axes explained 33.7% and 26.1% of genetic variance, respectively. The DAPC results supported our findings of high admixture between populations and identified ANF2, ANF3, and BRWMA2 as genetically distinct with limited inter-population gene flow (Figure 6). Populations SJSBP2 and THSF2 also appeared to be distinct, and to some extent LB1, ANF1, and ANF4, in a second DAPC (Appendix A) that removes the most divergent populations ANF2, ANF3, and BRWMA2. Therefore, the DAPCs results mirrored, in part, the sNMF analyses (Figure 5, Figure 6 and Appendix A).

### 2.3. Clonality

We identified clonal individuals in two 5 m diameter circular plots and identified a pair-wise genetic distance of 0.03 as the threshold below which individuals were considered to be clones (Appendix A). In the plot located in SJSBP, we identified five stems as belonging to one clone and the remaining 23 stems as belonging to another clone (CR = 0.04). The plot located in ANF contained two stems that were genetically distinct individuals and 19 stems belonging to one clone (CR = 0.10). Clones were spread across the entirety of the 5 m plots, and ramets of different genets were intermixed (Figure 7).

An analysis of the clonal diversity in the studied populations points out that 92% are multiclonal. There were one to 16 sampled clones in a single population (Figure 8, Table 2). The average clonal number per population was 8.4 ± 1.4 SE, and the average number of individuals per clone was 1.7 ± 0.23 SE. Clonal richness was the lowest in the LB and THSF1 populations (CR  =  0.09 and 0.08, respectively) and the highest in populations ANF 1, 2, 3, 4.3, and SJSBP 2 (CR = 0.82 to 1.0; Table 2). Similar to the circular plots, ramets of the same clone intermingled with other clones (Figure 8).

## 3. Discussion

We were able to describe genetic diversity and specifically assess the extent of clonal spread of *S. floridana* using next-generation sequencing, ddRADseq. Our finding that about 62% of the sampled plants were unique genotypes suggests that the *S. floridana* populations consist of a mixture of both asexually and sexually reproducing individuals. This demonstrates that sexual reproduction is occurring in most populations, more than was previously thought based on field observations [13]. Clonal propagation in some populations such as ANF5, LB, and THSF is high (CR ≤ 0.1), but fruit production has been observed over the years during surveys, suggesting the possibility of sexual reproduction.

*Scutellaria floridana* appears to possess moderately low genetic diversity and high levels of between-population differentiation in contrast to other members of the Lamiaceae of conservation concern [21,22]. Compared with a previous *S. floridana* study based on the AFLP marker [14], we observed lower estimates of expected heterozygosity (*H_E_* = 0.124–0.145 in this study versus *H_E_* = 0.145–0.184 in [14]) and higher levels of between-population differentiation (*S. floridana F*_ST_ = 0.035 in [14]). While differences in methodologies and markers used can make direct comparisons between the two studies difficult, the difference in *F*_ST_ estimates in particular may be due to the different sampling scheme and greater number of samples and areas included in this study. For example, the two populations that we identified as the most differentiated and least genetically diverse, BRWMA1 and ANF5, were not included in the previous work. This likely led to an underestimation of population differentiation and an overestimation of genetic diversity in the previous study.

Several populations displayed relatively large inbreeding coefficients and low observed heterozygosity compared with other Lamiaceae species and other *S. floridana* populations (Table 1) [21,22]. This could be due to low genet abundance, limited gene flow with other populations as a result of fragmentation, and decreased sexual reproduction, with clonal propagation dominating as a reproductive strategy [23,24,25]. Population BRWMA1, which possessed the lowest observed heterozygosity and highest inbreeding coefficient of all populations, is likely affected by all of these factors, as it displayed high divergence from other populations, a very low abundance of stems, no evidence of flowering during survey periods, and a clumped arrangement of stems across the landscape that were largely clonal (CR = 0.22, Figure 8). Population ANF5, which possessed similar observed heterozygosity values and inbreeding coefficients, also displayed high divergence and no evidence of flowering during our survey periods but had a much higher abundance of stems spread over a larger area. The sampling of ANF5 was of two subpopulations in close proximity, and the low clonal richness suggested that all sampled stems derived from a single genet (Figure 8). Theoretical models predict for long-lived, strictly clonal populations an increase in heterozygous loci, resulting in lower *F*_IS_ and *F*_ST_ [17,26]; however, the presence of even limited amounts of sexual reproduction can lead to increases in homozygosity and higher *F*_IS_ and *F*_ST_ [17]. It is possible that sexual reproduction among clonal individuals in these populations, which simulates selfing, could lower observed heterozygosity values and increases *F*_IS_ estimates.

We identified evidence of substantial clonal reproduction in our circular plots located in SJSBP2 and ANF4 (Figure 7). We also found variation in the number of clones found among the 14 populations and sites of *S. floridana* (Table 2), as well as in the spatial arrangement between ramets (Figure 8). The spatial pattern of clonal growth, where ramets are dispersed over considerable distances and intermixed with other clones across the landscape (Figure 8), can be characterized as the guerrilla type [15,27]. This strategy can increase geitonogamy, i.e., pollination between flowers of the same plant, leading to inbreeding depression. In contrast, the mixing of ramets of different clones can enhance cross-pollination. Therefore, according to our study, cross-pollination could be occurring in the populations in which the circular plots were established, since they possess moderately high genetic diversity relative to other populations and little evidence of inbreeding. It is worth noting that the mating system of *S. floridana* is currently unknown, but the possibility exists that this species exhibits both selfing and outcrossing, as has been documented for *S. angustifolia* complex [28] and *S. indica* in the form of dimorphic cleistogamy [29].

### Implications for Conservation and Management

The majority of *S. floridana*’s genetic diversity appears to be mostly encapsulated within a few populations that possess moderate heterozygosity, little or no evidence of inbreeding, unique ancestry, and high clonal richness (Table 1 and Table 2). Populations ANF2, ANF3, BRWMA2, and THSF2 fall into the highest category of conservation genetic value based on these criteria, and their continued persistence is vital to maintain the diversity, resiliency, and adaptive potential of *S. floridana* as a species. Populations ANF1, SJSBP2, LB1, and ANF4 fall into an intermediate category of conservation value and represent a significant portion of *S. floridana*’s reproductively viable individuals. Any of these seven populations would likely be suitable sources for ex situ conservation efforts. The remaining four populations, ANF5, BRWMA1, THSF1, and SJSBP1, displayed much lower levels of genetic diversity, moderate evidence of inbreeding, little unique ancestry, and low clonal richness. They were also moderately to highly diverged from most other populations, suggesting a lack of gene flow across the landscape. These populations should be further investigated to identify the cause of inbreeding and determine if management activities can stabilize or improve genetic diversity via the introduction of transplants from populations of high conservation value. Unfortunately, we do not know the level of inbreeding that is acceptable for these populations.

As our survey efforts did not reveal any individuals at the locations of the five previously documented populations on private lands that we were able to access, we suspect that they may be extirpated. Examination of historical survey data revealed that all of these populations had gone over 15 years without any monitoring efforts but were still assumed to be extant. Our findings suggest that monitoring efforts need to be significantly improved for populations of *S. floridana* that are located on private lands. We recommend that these populations be revisited and extensively surveyed to determine their status. If these five populations have been extirpated, the total number of extant populations of *S. floridana* would be reduced to 14, presenting evidence of a continued decline of the species since listing. Moreover, the geographic arrangement of the historically extirpated and suspected extirpated populations represents substantial fragmentation of the species’ range and increasing isolation of western populations.

Given how widespread clonal reproduction appears to be, it is likely that it plays a major role as a reproductive strategy. One possible factor driving clonal spread is fire, as clonal growth is common in disturbed habitats, and fire is an important disturbance for maintaining this species’ habitat and Florida ecosystems [30,31,32]. Furthermore, plants in fire-dependent ecosystems have shown positive responses after fire, such as increased resprouting and flowering [30,33,34,35]. *Scutellaria floridana* is locally abundant in areas managed with fire, such as the ANF and the SJBSBP, where flowering was extensive following recent burns [13]. *Scutellaria floridana* can be considered a resprouting species rather than a seeder, as the latter tends to regenerate solely by post-fire recruitment from a seed bank, and this species does not have a persistent seed bank [14]. Resprouting from below-ground tissue after fire is a key life-history trait of fire-dependent ecosystems, but the extent of *S. floridana* new ramet production and their lateral spread is currently unknown and needs to be investigated.

Our results also suggest that stem counts alone are an imperfect proxy for abundance in populations with low clonal richness, as one individual can be composed of dozens of stems with no clearly delineated shape or arrangement. Based on our circular plots, it appears that on average, ten adjacent stems generally represent 1–2 unique individuals, suggesting that stem counts are usually equivalent to ten times the true population size. This estimate may be even higher in small populations with low clonal richness. Our spatial analysis also reported that the populations on private lands that appear to be extirpated display noticeably different patterns of land cover than those located on public, managed lands. Areas on private lands that historically contained active populations of *S. floridana* contained less evergreen forest, which is generally described as ideal land cover for the species, and more low, medium, and high intensity development than the areas on public lands where populations have persisted [13]. *Scutellaria floridana* has the capability of prolonged vegetative dormancy, allowing it to persist in dense pine plantations; thus, one way to determine whether the species is not extirpated is to reintroduce a fire regime to those areas [13]. This suggests that *S. floridana* is heavily reliant on active, targeted land management with prescribed fire to maintain suitable habitat conditions and allow populations to persist over time.

## 4. Materials and Methods

### 4.1. Study Area

*Scutellaria floridana* is predominantly found in well-established longleaf pine flatwoods with a thin to moderate overstory of longleaf pine, an open understory with little to no shrub layer, and groundcover dominated by wiregrass [13] (Figure 1). This habitat occurs in frequently flooded lowland areas with elevations of approximately 0.5–10 m and sand and fine-sand soils. The annual mean temperature is 20 °C with an average high of 26 °C and a low of 15 °C, and the mean annual precipitation is approximately 147 cm [36]. *Scutellaria floridana* is endemic to the Florida Panhandle and is only documented in Liberty, Franklin, Gulf, and Bay counties (Figure 2). The majority of *S. floridana* populations are located on public lands with regular fire regimes that maintain suitable habitat conditions for growth and flowering.

The Florida Natural Areas Inventory (FNAI), Florida’s Natural Heritage Program and state member of the NatureServe network, has previously documented *S. floridana*’s range and identified all known occurrences of the species. Independent populations are defined as occurrences of the species that are at least 1 km away from the next nearest occurrence [37]. Because of this, populations are frequently composed of several spatially fragmented sub-populations, referred to in this paper as “sites.” This results in some very large populations composed of many sites and some much smaller populations containing only one site. We collected samples from at least one site in each population where *S. floridana* was found and collected from multiple sites within populations that were spread across a larger geographic area.

Of the 19 populations previously identified as extant, we were unable to access two due to fencing and an active timber harvest operation. All of the extant populations of *S. floridana* that we located in the field are found on managed lands. The majority of these are located along the western edge of ANF in management areas designated as longleaf pine and slash pine adaptive management units that are maintained by prescribed fire [38]. Other populations were located in THSF, BRWMA, SJSBP, and LB (Figure 2). Tate’s Hell State Forest is located directly south of ANF in the lower coastal plain along the coast of the Gulf of Mexico and is managed by prescribed fire with a burn frequency of 2–15 years [39]. Box-R Wildlife Management Area is located on the Gulf coast approximately 15 km west of THSF and is managed with selective thinning and prescribed fire with a burn frequency of 1–5 years [40]. St. Joseph Bay State Buffer Preserve is situated on the Gulf coast approximately 10 km west of BRWMA. Relevant areas that contain *S. floridana* are designated as wet prairie and are managed with prescribed fire at a 2–3 year frequency [41]. Lathrop Bayou is located approximately 30 km west of ANF and includes four islands at the eastern end of East Bay. *Scutellaria floridana* is found on the largest of these islands, Raffield Island. The primary management goal of this area has been to restore open-understory pine flatwood conditions using prescribed burns at a frequency of 1–2 years [42].

### 4.2. Tissue Collection and Storage

We conducted two trips in the spring of 2021 and collected a total of 294 plant tissue samples from 12 populations, representing 17 sites (Table 1). We took samples from individuals separated by at least one meter since the sites were generally patchy. Sampling sites such as ANF5 show a diagonal distribution as plants were found near a wetland (Figure 8). We georeferenced each sample collected using a Bad Elf GNSS Surveyor (Bad Elf West Hartford, CT). Plant collections were permitted under the Florida Department of Agriculture and Consumer Services Division of Plant Industry (#2021-03-002), US Department of Agriculture Forest Service Special Use Permit (WAK04122020), US Department of Interior Fish and Wildlife Service 10(a)(1)(A) permit to VNO, and Florida Fish and Wildlife Conservation Commission (SUO-82149).

For our analysis of genetic diversity, we conducted wide-scale sampling of 10–15 individual stems across each site where *S. floridana* was present, with a minimum distance of approximately 1 m between each sample (Table 1). For our analysis of clonality, we conducted intensive sampling within circular plots of 5 m in diameter, with a randomly selected midpoint placed in an area of high stem density. We established two plots in the ANF and one plot in the SJSBP and collected 30–40 stems per plot. To ensure accurate spatial analysis, we recorded distance (cm) and azimuth to the center of the plot for each sample using a compass and tape measure and manually placed sample locations onto a map using ArcGIS Pro (v. 2.7, ESRI 2011, Redlands, CA: Environmental Systems Research Institute). For each sample, we collected the entirety of the above-ground stem (including stem, leaves, and flowers) to ensure enough tissue for DNA isolation. Samples were stored at 4 °C after collection for up to 5 days, then transferred to −80 °C upon return to Miami University, OH. Voucher-photographed specimens from select populations were deposited in the Miami University (MU) Herbarium.

### 4.3. DNA Isolation and Sequencing

We extracted total genomic DNA from our plant material samples using a protocol similar to that used by [43]. Briefly, frozen samples were finely ground in liquid N_2_ and dissolved in an extraction buffer containing 100mM Tris, pH 8.0, 50 mM EDTA, 500 mM NaCl, and 0.1% W:V PVP 40, followed by 5M potassium acetate precipitation of cellular debris and isopropanol precipitation of genomic DNA. We assessed the quality of the DNA from the samples using gel electrophoresis on a 1.5% agarose gel in Tris-Acetate-EDTA buffer to ensure there was little to no DNA degradation. We estimated the quantity of DNA in our samples using a Qubit 4 fluorometer (Thermo Fisher Scientific, Waltham, MA, USA). Samples that displayed adequate quality and reached a minimum DNA concentration of 20 ng/uL were then sent to Floragenex (Floragenex, Inc., 4640 SW Macadam Ave., Portland, OR), where double-digest restriction site-associated DNA sequencing (ddRAD-Seq) was carried out. To summarize, DNA was first digested using the restriction endonucleases PstI and MseI. Samples were diluted for PCR amplification, and the product was used to construct a ddRAD-Seq library. The library was sequenced at the University of Oregon Genomics and Cell Characterization Core Facility (GC3F) on a NovaSeq 6000 with a SP100 chip, generating 118 bp single-end reads with a mean 27.5× effective coverage per sample. The sequence data were run through the pipeline STACKS (version 2.60) to assemble the short-read sequences from all the samples (via the process radtags program) and to align reads into loci that are genotyped (via the gstacks program) [44,45]. Single nucleotide polymorphism data were exported in VCF version 4.2 file format for downstream data analysis (see below). Three quality cut-off filters were applied, allowing for genotypes to be present in 40%, 60%, or 80% of individuals. The final dataset used maximized the number of variable sites while keeping the proportion of missing data per site at 40% or lower. The 80% presence cutoff dataset was not used as the number of loci was reduced to 227 and the number of variable sites to 102.

### 4.4. Spatial Analysis

We analyzed changes in habitat over the past 20 years using 2001 and 2019 land cover rasters retrieved from the USGS National Land Cover Database. We used ArcGIS Pro (Esri ArcGIS Pro v. 2.9) to calculate the total area and change in area of each land cover category over the past 20 years across the entirety of *S. floridana*’s range, on managed lands within its range, and within 500 m of each population.

### 4.5. Genetic Diversity Assessment

Floragenex filtered raw sequence data and identified and assembled loci using the Stacks pipeline [44]. We used a dataset in which each locus was represented by at least 60% of individuals; datasets with less missing data (found in 80% of individuals) resulted in a loss of informative loci. To assess within-population genetic diversity, we calculated heterozygosity and inbreeding coefficients for each population using the R package hierfstat [46]. To assess genetic differentiation between populations, we calculated pairwise *F*_ST_ for populations using the package StaMPP [47]. To investigate isolation by distance, we ran a Mantel test for a significant relationship between pairwise *F*_ST_ and geographic distance between populations using the package ade4 [48]. We estimated ancestry coefficients for individuals via an sNMF analysis using the package LEA [49] and performed a discriminate analysis of principal components (DAPC) using the R package hierfstat. We performed cross-validation to determine the optimal number of principal components (PCs) to retain.

The population genetic value was qualitatively assessed based on the level of a population’s observed heterozygosity, inbreeding coefficient, population divergence (based on *F*_ST_ and DAPC analyses), and signature of unique genetic ancestry (based on sNMF analysis) relative to other populations in this study. High to very high genetic value populations had high estimates of observed heterozygosity and low inbreeding coefficients, high or very high estimates of population divergence (based on *F*st and DAPC analyses), and signatures of unique ancestry. Moderate genetic value populations have moderate estimates of observed heterozygosity and inbreeding coefficients, moderate estimates of population divergence, and exhibit signatures of unique ancestry. Finally, low genetic value populations have low estimates of observed heterozygosity, high inbreeding coefficients, moderate estimates of population divergence, and lack signatures of unique ancestry.

### 4.6. Clonal Assignment and Analyses

We identified clonal individuals from two circular plots and from 12 populations sampled across the species range using the package poppr version 2.9.3 [50]. The clonal analysis for ANF4 was divided among three sampling sites because of its broad spatial distribution. We set the threshold to distinguish unique genotypes from clones by generating frequency histograms of genetic distance between samples and identifying the location of a large gap between values (Appendix A). We then constructed UPGMA trees and identified clones as any group of individuals that diverged to the right of the distance value corresponding to this threshold (Appendix A). For the circular plots, we filtered individuals missing more than 25% of SNPs from the plot in SJSBP2 and individuals missing >45% of SNPs from one plot in ANF4, as this method was sensitive to missing data. Because over half of the individuals in our second circular plot located in ANF4 were missing significant amounts of data, we excluded this plot from analysis. Our distance threshold to distinguish individuals was set at 0.03 for both plots. For population clonal analysis, we constrained the ploidy level to diploid before running the analysis in poppr. We identified the number of clones (genets) and the number of individuals per clone (ramets) in each population or site. Clonal richness (CR) was calculated as the number of genotypes (G) relative to the number (N) of samples assessed (CR = (G-1)/(N-1); [51]). The spatial arrangement of the samples assigned to the corresponding clones was visualized using their longitude and latitude coordinates; however, given the classification status of this species, we cannot report the specific coordinates.

## 5. Conclusions

Overall, *S. floridana* appears to have suffered a continued decline since its listing as threatened under the ESA, and populations seem to be heavily reliant on management activities to remain extant. The 12 populations that we were able to confirm as extant possess low genetic diversity, and four of them present considerable evidence of inbreeding. We recommend thorough surveying of populations located on private lands to determine whether they are still extant and an in-depth investigation of clonal reproduction and management history of the four populations on public lands (BRWMA1, ANF5, THSF1, and SJSBP1) that possess very low genetic diversity and evidence of inbreeding. If the high prevalence of clonality is driving the low genetic diversity in these populations, re-establishing disturbance regimes, especially prescribed fire, could stimulate flowering and potentially sexual reproduction and improve genetic diversity [13]. We identified four populations (ANF2, ANF3, BRWMA2, and THSF2) that possess most of the genetic diversity of *S. floridana* as a species and suggest that they be treated as the highest conservation priority, as their continued existence is vital to preserve the adaptive potential of the species.

*Scutellaria floridana* will likely require regular monitoring and active conservation efforts, including public outreach, to avoid any further extirpation events. There is a possibility, however, that the *S. floridana* genotypes will persist in a population longer, since the guerrilla strategy tends to reduce the chance of extirpation, lessening the impact of stochastic events on the genetic structure of this species. The combination of sexual and asexual (in the form of clonal growth) reproduction may be advantageous for maintaining the viability of *S. floridana* populations.

## Figures and Tables

**Figure 1 plants-12-00919-f001:**
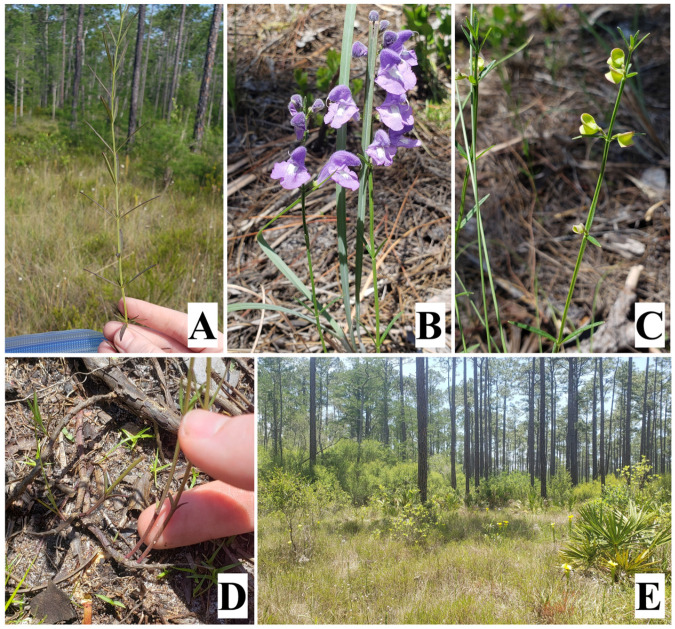
*Scutellaria floridana* plants, flowers, fruits, and habitat. (**A**) vegetative growth, ANF, (**B**) flowers, BRWMA, (**C**) fruits, BRWMA, (**D**) rhizome, BRWMA, and (**E**) typical habitat of *S. floridana*, ANF.

**Figure 2 plants-12-00919-f002:**
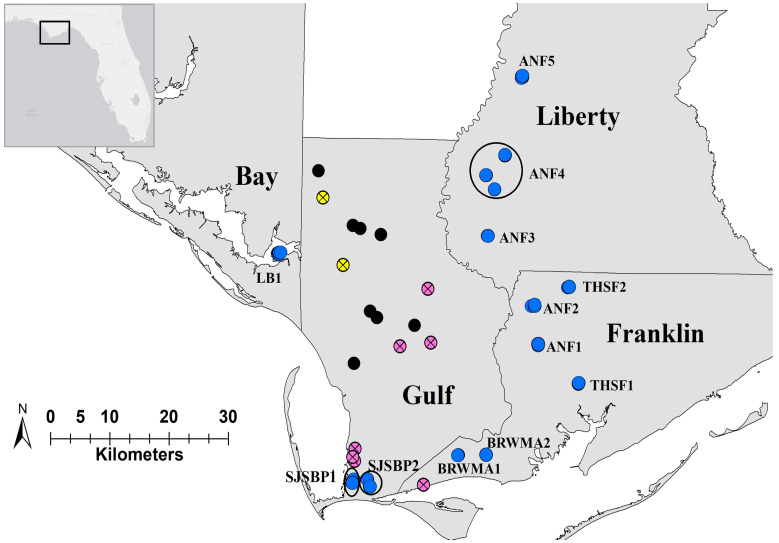
Locations of known occurrences of *S. floridana*. Black circles represent officially extirpated sites, yellow circles represent sites that were inaccessible at the time of our field surveys, pink circles represent sites in which we were unable to locate any individuals during field surveys, and blue circles represent populations that we successfully located and collected samples from for analysis of genetic diversity. Field surveys and sample collection were carried out in March and May of 2021. The Apalachicola River follows the boundaries of Gulf, Franklin, and Liberty counties.

**Figure 3 plants-12-00919-f003:**
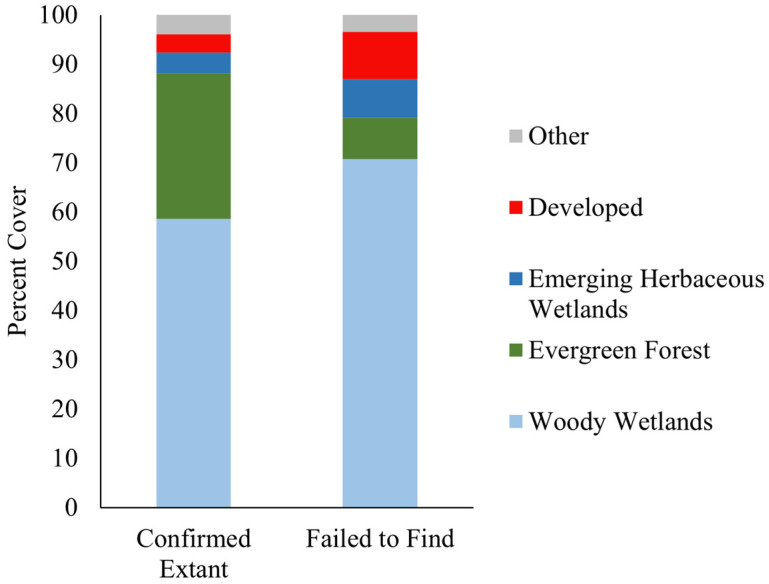
Percent cover of dominant land cover categories within 500 m of *S. floridana* populations that were confirmed to be extant and populations in which we were unable to locate any individuals during 2021 field surveys. See Appendix A for more information.

**Figure 4 plants-12-00919-f004:**
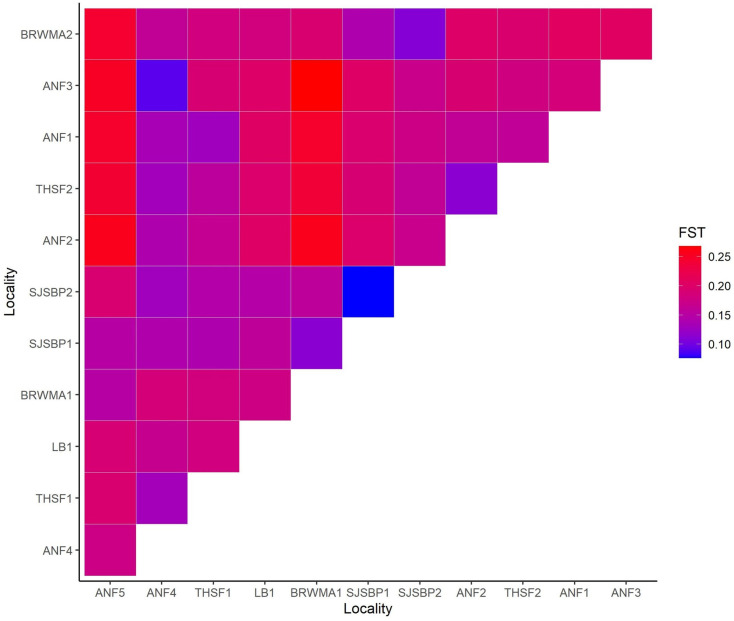
Heatmap visualizing pairwise *F*_ST_ values for twelve populations across *S. floridana*’s range.

**Figure 5 plants-12-00919-f005:**
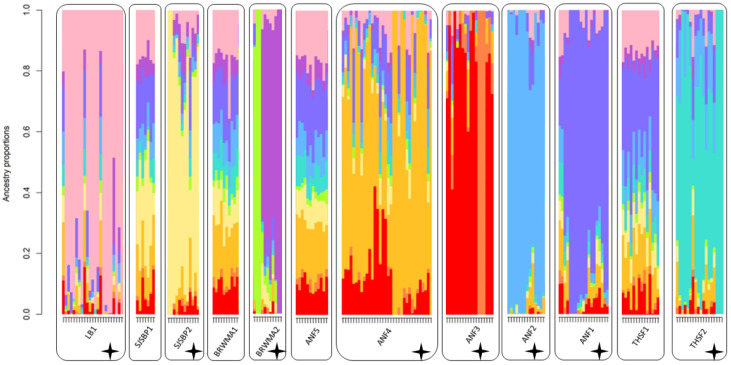
Ancestry proportions of individuals collected from twelve populations across *S. floridana*’s range (K = 10) based on sNMF analysis. Individuals are grouped by population (LB = Lathrop Bayou; SJSBP = St. Joseph Bay State Buffer Preserve; BRWMA = Box-R Wildlife Management Area; ANF = Apalachicola National Forest; and THSF = Tate’s Hell State Forest). Populations with unique ancestry are marked with asterisks. Colors indicate different genetic ancestry within individuals.

**Figure 6 plants-12-00919-f006:**
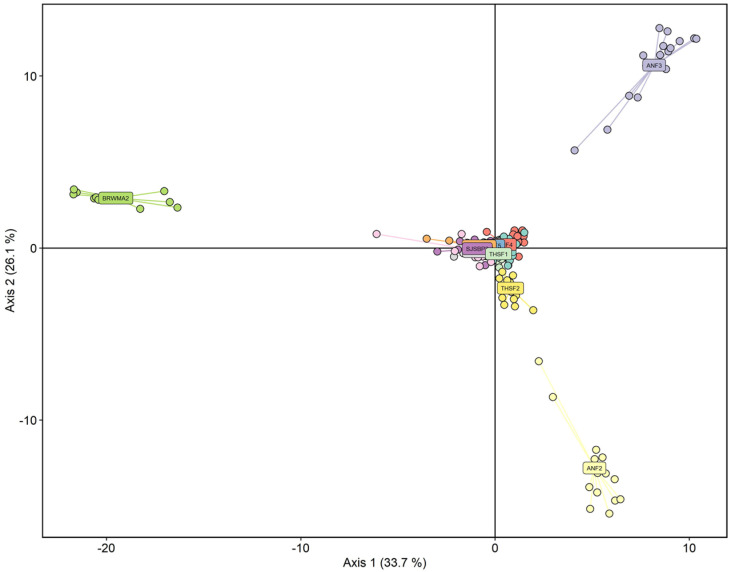
Discriminate analysis of principal components (DAPC) of twelve populations of *S. floridana* using 20 principal components and eight axes. Populations are shown with different colors, circles represent individuals, and labels are placed on the centroid (average) position for each population.

**Figure 7 plants-12-00919-f007:**
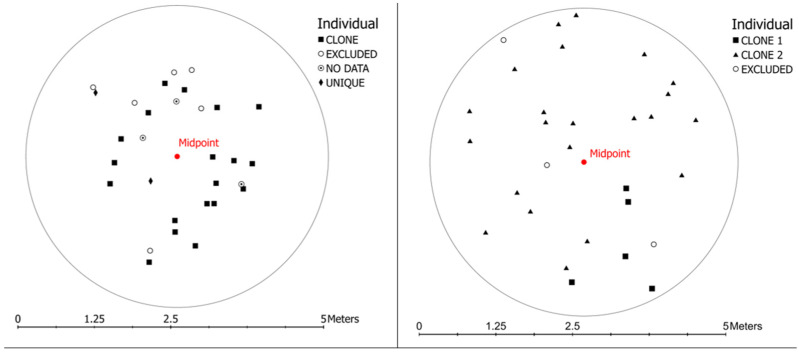
*Scutellaria floridana* clone identification (squares and triangles) and genetically unique individuals (diamonds) in 5m circular plots established in ANF4 (**left**, N = 21) and SJSBP2 (**right**, N = 28). Each plot was 5 m in diameter, with a midpoint selected randomly in an area displaying high stem density. Excluded individuals (open circles) in each plot were missing 45% (ANF4) or 25% (SJSBP2) of SNPS. Individuals with no data (circle with dot) were excluded from analyses.

**Figure 8 plants-12-00919-f008:**
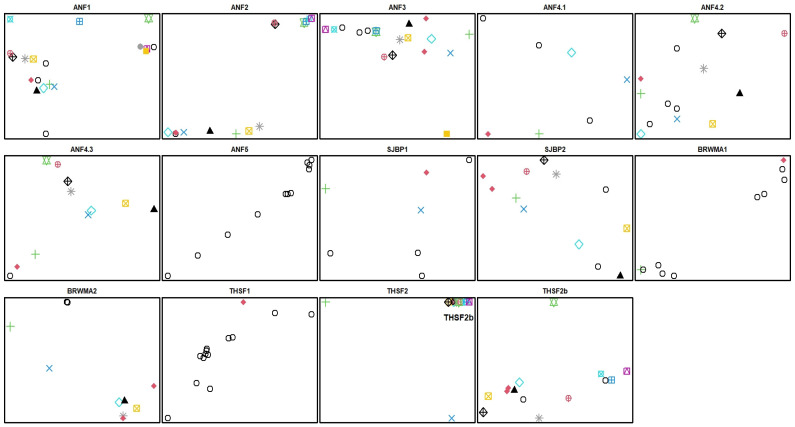
Spatial distribution of *S. floridana* clones and individuals with unique profiles within studied populations (see Figure 2 for locations) based on longitude (*x*-axis) and latitude (*y*-axis). Symbols represent each sampled individual. Within each population, ramets belonging to the same clone possess the same symbol. The last panel, THSF2b, shows the spatial distribution of a subset of THSF2 data. ANF4 was divided among three sampling sites because of its broad spatial distribution.

**Table 1 plants-12-00919-t001:** Locations, number of sites, and samples collected; expected (*H_E_*) and observed (*H_O_*) heterozygosity; inbreeding coefficient (*F*_IS_); relative evidence of population divergence (based on *F*_ST_ and DAPC); evidence of unique ancestry; and relative genetic value of twelve populations of *S. floridana*. Samples from LB were collected by Ms. Amy Jenkins (FNAI) and mailed to Miami University, OH, in April 2020. Genetic value determined as described in Materials and Methods.

County	Population	Sites	*N*		*H_E_*	*H_O_*	*F* _IS_	Population Divergence	Unique Ancestry	Genetic Value
Bay	LB1	1	23		0.142	0.125	0.113	Moderate	Yes	Moderate
Franklin	ANF1	1	19		0.125	0.106	0.137	Moderate	Yes	Moderate
Franklin	ANF2	1	14		0.128	0.127	0.024	Very High	Yes	High
Franklin	BRWMA1	1	10		0.139	0.04	0.555	Moderate	No	Low
Franklin	BRWMA2	1	11		0.136	0.139	−0.01	Very High	Yes	High
Franklin	THSF1	1	14		0.131	0.081	0.297	Moderate	No	Low
Franklin	THSF2	2	18		0.131	0.13	0.025	High	Yes	High
Gulf	SJSBP1	2	7		0.145	0.090	0.268	Moderate	No	Low
Gulf	SJSBP2	1	12		0.145	0.136	0.060	Moderate	Yes	Moderate
Liberty	ANF3	1	18		0.126	0.132	−0.02	Very High	Yes	High
Liberty	ANF4	3	34		0.138	0.116	0.154	Low	Yes	Moderate
Liberty	ANF5	2	11		0.124	0.044	0.504	Moderate	No	Low
	Total:	17	191	Mean:	0.134	0.106	0.176			

**Table 2 plants-12-00919-t002:** Estimation of the number of unique clones (genotypes), ramets, and clonal richness per *S. floridana* population. N = sample size. Fifth column reports the number of ramets per clone (frequency in parentheses). CR = clonal richness, CR = (G − 1)/(N − 1) (49). Clonal analysis for ANF4 was divided among three sampling sites because of its broad spatial distribution. n/a = no gap in the histogram to allow for a threshold designation for clones; for these populations, the minimum pairwise diversity was ≥ 0.04.

Population	N	Clonal Diversity Threshold	Number of Unique Clones	Number of Ramets per Unique Clone	CR
ANF1	19	0.01	16	4, 1 (15)	0.83
ANF2	14	n/a	14	1 (14)	1.00
ANF3	18	0.02	15	3, 2, 1 (13)	0.82
ANF4.1	7	0.01	5	3, 1 (4)	0.67
ANF4.2	15	0.01	11	5, 1 (10)	0.71
ANF4.3	11	n/a	11	1 (11)	1.00
ANF5	11	0.02	1	11(1)	0
BRWMA1	10	0.002	3	8, 1 (2)	0.22
BRWMA2	11	0.02	8	3, 2, 1 (6)	0.70
LB1	23	0.02	3	18, 3, 2	0.09
SJSBP1	7	0.01	4	4, 1 (3)	0.50
SJSBP2	12	0.03	10	2, 2, 1 (8)	0.82
THSF1	14	0.03	2	13, 1	0.08
THSF2	18	0.01	14	3, 2, 2, 1 (11)	0.76
Total, or Average ± standard error	190		117	1.7 ± 0.23	0.59 ± 0.09

## Data Availability

We uploaded SNP data and vcf files to Dryad.

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
