# Peer review of "High Prevalence of Clonal Reproduction and Low Genetic Diversity in Scutellaria floridana, a Federally Threatened Florida-Endemic Mint"

_plants, 2023, doi:10.3390/plants12040919_

Round 1
Reviewer 1 Report
Overall, Hanko et al provide a nice study of an endangered plant species using good sampling and appropriate genetic markers. The methods are sound and the paper generally well presented. There can help improve the paper noted below.
Probably good to add the Apalachicola River to the map and maybe in the discussion mention this major biogeographical boundary as noted in Soltis et al 2006 DOI: 10.1111/j.1365-294X.2006.03061.x
I wonder how the results are influenced by the finding that ANF5 consists of a single clone. Seems interesting that this population has at least 2 distinct sites and 11 samples were collected, yet they all seem to be a single clone (Table 2). How does that play into the Fst findings?
Following up on that, it seems a bit counter-intuitive that there’s a high Fst between ANF5 and other populations while the DAPC shows it right in the middle of things…Any ideas on that?
Ln 18: “we detected low genetic diversity (HE= 0.125-0.145)…”—Relative to what?? What makes He 0.125 low? “… moderate evidence of inbreeding (FIS = -0.02-0.555),…” FIS=0 is random mating, not sure this is accurate. Some populations show higher levels, but FIS=-0.02 is not “moderate evidence of inbreeding”.
Ln 151—table 1: How is this table sorted? Seems like populations from the same management unit (BRWMA1 and BRWMA2) should be consecutive lines rather than either end of the block for Frankiln co.
Ln 155-161: Without the populations labeled on the map, it’s hard to interpret the results here…where are BRWMA1 and ANF5 relative to each other?
Ln 214—Fig 8: Might be nice to add general outlines of the populations if available. e.g. is ANF5 a linear population running diagonally? Or is that how it was sampled? Is the sampling in THSF2 clumped in the NE because that’s where the plants are? Were “clones” ever found in different populations?
Ln 285-286: Are the land owners conservation oriented? Unfortunately, many private land owners in Florida have been known to intentionally destroy populations of rare plants in fear that their land will be seized. Unless more is known, it may not be wise to do much with the private lands…
Ln 316: italicize Pinus palustris
Ln 317-318: italicize Aristida strict
Ln 327-fig1 legend: Italicize spp names
Ln 339: Fig 2: Label populations
Ln 405: Here or below in section 4.5, you need more information on STACKS: version, settings, etc.
Ln 426: Population genetic value seems a bit made-up. Also given that you have 4 measures going into this, how were each weighted? Low vs high relative to what?
Ln 429—noticed here, but throughout the manuscript need to be consistent with the use of FST, Fst or likely F\sub(ST)
Reviewer 2 Report
This is a good case study showing how genetic data can and should be evaluated in conservation biology. It contributes in raising awareness on the importance of including genes in conservation and gives applied recommendation in future conservation needs.
The introduction reads well. In page 1, line 44, it’s stated that “plants are specially…” following an argument that they have limited migration capabilities and adaptations to soil, etc. Not all plants will be negatively affected by climate change, so I suggest this is modified to rare plants, as seems to be the topic of the paragraph.
The analyses of land cover are not presented in the aims. It takes up quite a large part of methods and results and is used in the Discussion. I like how the authors have used this information to draw conclusions on the conservation needs. It should be stated as a part of the aims.
In page 3, autor name of S. stricum is missing.
Page 3, line 111, 113-114. Abbreviations must be written out first time.
Page 5, line 159. Why is this interesting? Please use another word, eg. In fact, … or something, and make the text interesting to the reader instead of stating it.
Figure 5. Are there supposed to be asterisks in the figure? I can’t see any. And, what happens if you lower the K to eg. 4? Does it fit with the results of figure 6?
Figure 6. No need for labels when you have a legend. They overlap anyway.
Page 8, line 204. Delete ranged.
Page 8, lines 205-206. Move “population” to before population names are listed.
The Discussion gives recommendations for conservation in line with findings that is useful for decision makers. I only have minor input on the Discussion.
Page 9, line 224. I miss a reference to the statement: …, more than was previously thought.
Page 9, line 230. Could molecular methods used give different results in observed heterozygosity.
Page 10, lines 265-267. Is TSHF2 enough to represent the cluster of populations in Figure 6? It seems to be a bit separated from the rest.
Page 10, line 276-278. Is inbreeding a problem? Are there reasons to believe that it will affect the populations negatively? Since mating system is unknow, it might be at an acceptable level.
Page 10, lines 295-298. Is there a seed bank? It is not clear to me how the plants respond to fire. Is the lack of fire also the potential reason why populations in private land seems to be gone? This can be stated clearer, if that is the case.
Materials and methods. Species names in first paragraph have to be in italics.
Figure 2. Please change color of orange and red. They appear as yellow and orange to me and reading the figure text is thus confusing. Also, avoid red and green in the same figure.
Page 14, lines 400-406. This is very shortly summarized, please extend on the methods used or refer to studies with the same setup.
Page 15, line 433. Delete one have.
Page 15, line 456. Delete one .
Page 15, line 458. What listing? Please remind the reader.
